# Intensity Correction and Standardization for Electron Microscopy Data

**Oleh Dzyubachyk**[1,2]                                   O.DZYUBACHYK@LUMC.NL
**Roman I. Koning**[1]                                       R.I.KONING@LUMC.NL
**Aat A. Mulder**[1]                                          A.A.MULDER@LUMC.NL
**M. Cristina Avramut**[1]                              M.C.AVRAMUT@LUMC.NL
**Frank G.A. Faas**[1]                                       F.G.A.FAAS@LUMC.NL
**Abraham J. Koster**[1]                                   A.J.KOSTER@LUMC.NL

[1] *Section Electron Microscopy, Department of Cell and Chemical Biology, Leiden University Medical Center, Leiden, the Netherlands*
[2] *Division of Image Processing, Department of Radiology, Leiden University Medical Center, Leiden, the Netherlands*

## Abstract

Intensity of acquired electron microscopy data is subjected to large variability due to the interplay of many different factors, such as microscope and camera settings used for data acquisition, sample thickness, specimen staining protocol and more. In this work, we developed an efficient method for performing intensity inhomogeneity correction on a single set of combined transmission electron microscopy (TEM) images and demonstrated its positive impact on training a neural network on these data. In addition, we investigated what impact different intensity standardization methods have on the training performance, both for data originating from a single source as well as from several different sources. As a concrete example, we considered the problem of segmenting mitochondria from EM data and demonstrated that we were able to obtain promising results when training our network on a large array of highly-variable in-house TEM data.

**Keywords:** Transmission electron microscopy, Mitochondria segmentation, Intensity correction, Intensity standardization

## 1. Introduction

Even though quantification of electron microscopy (EM) data has received a significant boost with the advance of machine-learning techniques, the number of publications on this type of imaging still lags greatly behind other modalities. In particular, the work of Lucchi et al. (2013) was one of the first publications on machine-learning-based segmentation of mitochondria from EM data that received significant attention in the community. The data that were made publicly available by the authors had become a benchmark for virtually all subsequent studies on this topic. Other publications that attracted large attention in this area are the works of Haberl et al. (2018) and Xiao et al. (2018). In particular, the latter publication described an elaborate network design for segmentation of mitochondria from EM data that served as a basis for follow-up publications of other groups (Casser et al., 2020).

While network design has received significant attention in the literature, the problem of standardization (harmonization) of EM data remains largely unaddressed. The only type of data preprocessing we are able to find in the related publications was histogram

equalization (Xiao et al., 2018), but even in this case no further implementation details were provided. This fact might be attributed to rather small data size on which these methods were trained and tested. However, our in-house data set is much larger: hundreds of data sets, each consisting of several hundred or even several thousand of separate frames. This inevitably leads to significant variability of the intensity distribution of these images, which, in combination with large diversity in the appearance of the mitochondria themselves due to biological variations and staining properties (see Section 2), renders quantification of such data extremely challenging.

The aim of this research is to develop targeted preprocessing methods for EM data, in particular, intensity correction and standardization, with the final goal of developing a machine-learning-based segmentation approach for processing a wide variety of EM images. As a concrete example, in this work we considered the problem of automated segmentation of mitochondria. However, we expect the performed analysis and developed approaches to be generic and to have capacity to be extended to a wide range of similar problems, not limited to EM data. In the remainder of this manuscript, we present our experiments and draw several important conclusions from the results.

## 2. Data

For this analysis, we selected several image data sets from our in-house data acquired as part of the same project. The images were acquired with a digital charge coupled (CCD) camera (One View, Gatan Inc., Pleasanton, USA) mounted on a Tecnai 12 TWIN transmission electron microscope (FEI, Eindhoven, the Netherlands) operating at 120 kV. CCD images of fixed and positively stained samples (Giacomelli et al., 2020) were collected with binning 2 and an overlap of 20% and stitched together into one large image mosaic, as described previously (Faas et al., 2012). The samples for acquiring all the data analyzed in this project originated from the kidney tissue of 3 different individuals (donors) and were imaged at a single (one donor) or two different time points (two donors).

Each image mosaic typically consists of several hundred of separate frames, $2048 \times 2048$ pixels large (pixel size $= 3.35$ nm$^2$). Three experts on this type of EM images have selected regions of interest (ROI's) from five mosaics and manually segmented mitochondria on the original frames belonging to these ROI's using custom in-house annotation software. Each data set was annotated by one expert. Statistics on annotated frames and total number of annotated objects are provided in Table 1. A typical frame from each data set and the corresponding mitochondria annotations are shown in Figure 1. These images confirm significant variability of mitochondria appearance with respect to size, shape and intensity distribution. As we are using exactly the same preparation and acquisition protocol for all these data sets, the observed differences are entirely caused by the underlying biological factors. Quantitative analysis of these differences is the main objective of the large research project, with this work being the first step towards achieving this goal. A contiguous $10 \times 10$ frames region was additionally selected from each of the annotated data sets, close to the location of the annotated ROI, for developing and testing the intensity inhomogeneity correction algorithm.

Table 1: Number of annotated frames and mitochondria from each selected data set

|  | 2922Q1 | 2922Q4 | 2929L4 | 2929Q1 | 2929Q4 |
|---|---|---|---|---|---|
| Frames | 20 | 44 | 58 | 251 | 55 |
| Mitochondria | 222 | 598 | 1927 | 2764 | 745 |

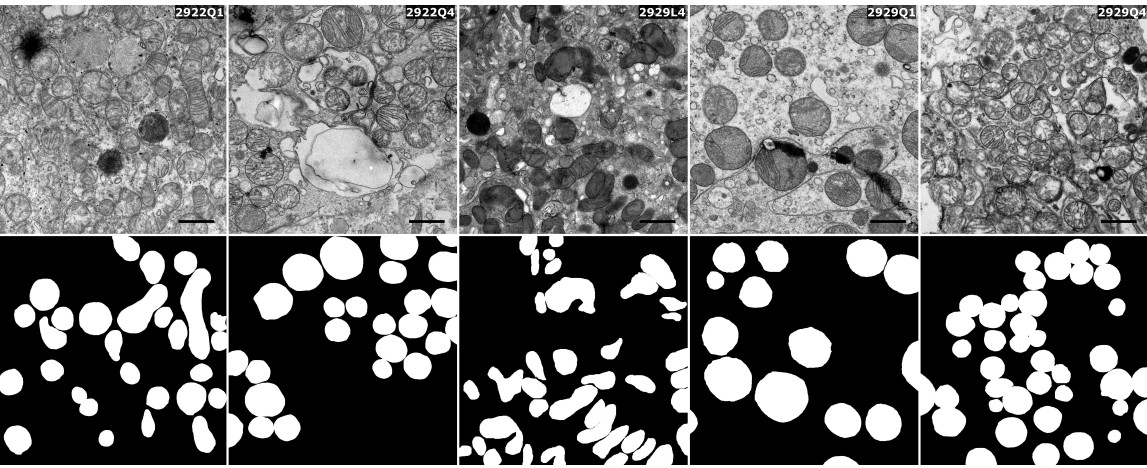

Figure 1: Sample frame from each data set (top) and the corresponding annotation (bottom). Contrast of the images in the top row was increased for visualization purposes. The length of the scale bar is 1 μm.

## 3. Experiments and Results

In this section, we present setups and results of three experiments designed to assess data quality improvement after applying intensity correction and standardization on the selected EM data. The first experiment was executed on the non-annotated data originating from the five data sets, and the remaining two experiments were executed on the annotated data. In the remainder of this manuscript, we will use following abbreviations for different intensity correction and standardization methods: "B" = bias correction, "H" = histogram equalization, "M" = histogram mapping.

To train a deep convolutional network, we adopted the design of Xiao et al. (2018). However, our architecture was simpler as we did not use some of the advanced features, such as auxiliary outputs or augmentation during test phase. The dropout rate was set to 0.1, the batch size was set to 4, and we used geometric data augmentation (flipping, rotation by 90°, 180° and 270°) at the training phase. The network was trained for 50 epochs, using the weighted sum of the binary cross-entropy and the Jaccard index as the loss function. All frames were downsampled to the size of $256 \times 256$ pixels prior to training. For each data set, 15% of the annotated frames were reserved for testing, 15% — for validation, and the remaining 70% were used for training. For each repetition of the training experiment, we generated a separate data split using different random seeds. Consequently, we processed these data with each of the described preprocessing methods, in turn.

### 3.1. Intra-set intensity correction

The goal of this experiment is to develop an approach for reducing intensity variation within a single data set. Such a method should potentially be able to perform both intensity scaling and bias (field inhomogeneity) correction. For this, we extended the Coherent Local Intensity Clustering (CLIC) method (Li et al., 2009) that was developed for correcting magnetic resonance data. To access the quality of intensity correction, we used information from the corresponding overlapping regions of two neighbouring frames of five test data sets. More precisely, we selected the absolute difference between the means of the two overlapping regions and the Jeffrey divergence (Jäger and Hornegger, 2009) between the histograms of these regions as our validation measures.

CLIC (Li et al., 2009) is an elegant framework that allows performing intensity inhomogeneity correction based on a very limited set of assumptions that: 1) intensity content of every frame is modelled as a combination of a finite number of classes, each having a distinct intensity distribution; and 2) the bias field is smooth. Following their formalism and adding novel linear intensity correction terms (shift $\mathbf{a}$ and scaling $\mathbf{b}$) to the model, we represent every acquired frame $I_t(\mathbf{x})$ ($t = \overline{1, N}$) as:

$$I_t(\mathbf{x}) = a_t + b_t B(\mathbf{x}) J_t(\mathbf{x}) + n_t(\mathbf{x}),$$

where $J_t(\mathbf{x})$ is the true intensity of the frame; $B(\mathbf{x})$ is a smooth bias field; $\mathbf{b} = \{b_t\}$ and $\mathbf{a} = \{a_t\}$ are the slope and the intercept of the linear intensity correction function; $n_t(\mathbf{x})$ denotes additive noise; $N$ is the total number of frames; and $\mathbf{x}$ is a vector of 2D Cartesian coordinates. Note that the bias field is assumed to be the same for all frames as it should represent imperfection of the imaging device. Conversely, the slope $a_t$ and the intercept $b_t$ of the intensity correction function, modelling intensity shift and scaling, respectively, are constant for every frame $I_t(\mathbf{x})$.

Denoting the target intensity of each of the three classes observed as peaks on the intensity histogram of the data set as $\mathbf{c} = \{c_i\}$ ($i = \overline{1, 3}$), we arrive at the following energy function:

$$\mathcal{J}_{\mathbf{x}}^{loc}(U, \mathbf{a}, \mathbf{b}, \mathbf{c}, B) \triangleq \sum_{t=1}^{N} \sum_{i=1}^{3} \int_{\mathcal{O}_{\mathbf{x}}} u_{t;i}^q(\mathbf{y}) K(\mathbf{x} - \mathbf{y}) \left| I_t(\mathbf{y}) - a_t - b_t c_i B(\mathbf{x}) \right| d\mathbf{y}.$$

Here $q$ is a real weight (we used $q = 2$ in all our experiments); $U = \{u_{t;i}(\mathbf{x})\}$ is the class membership function defining probability of each particular pixel belonging to the corresponding intensity class and $K(\mathbf{x})$ is the truncated Gaussian kernel defined on the neighbourhood $\mathcal{O}_{\mathbf{x}}$. For the strict definition of these parameters and a more detailed explanation about them we refer the readers to the original publication by Li et al. (2009).

Minimizing $\mathcal{J}^{loc}$ with respect to the variables $\mathbf{a}$, $\mathbf{b}$ and $B$ results in the following equations for the corresponding parameters:

$$a_{t=\overline{1,N}} = \frac{\displaystyle\sum_{i=1}^{3} \int_{\mathcal{O}_{\mathbf{x}}} u_{t;i}^q(\mathbf{x}) I_t(\mathbf{x}) \left[ c_i (K * B(\mathbf{x})) - b_t I_t(\mathbf{x}) \right] d\mathbf{x}}{\displaystyle\sum_{i=1}^{3} \int_{\mathcal{O}_{\mathbf{x}}} u_{t;i}^q(\mathbf{x}) d\mathbf{x}},$$

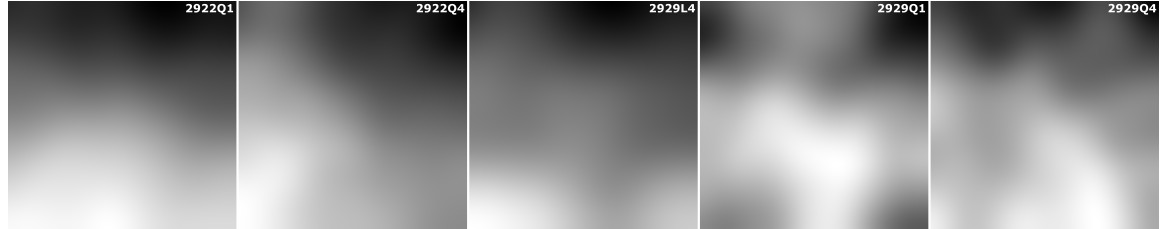

Figure 2: Estimated intensity inhomogeneity (bias) field for each of the five data sets.

$$b_{t=\overline{1,N}} = \frac{\sum\limits_{i=1}^{3} \int_{\mathcal{O}_{\mathbf{x}}} u_{t;i}^q(\mathbf{x}) I_t(\mathbf{x}) \left[ c_i(K * B(\mathbf{x})) - a_t \right] d\mathbf{x}}{\sum\limits_{i=1}^{3} \int_{\mathcal{O}_{\mathbf{x}}} u_{t;i}^q(\mathbf{x}) I_t^2(\mathbf{x}) d\mathbf{x}}, \tag{1}$$

$$B = \frac{K * \left( \sum\limits_{t=1}^{N} \sum\limits_{i=1}^{3} \int_{\mathcal{O}_{\mathbf{x}}} u_{t;i}^q(\mathbf{x}) c_i \left[ a_t + b_t I_t(\mathbf{x}) \right] d\mathbf{x} \right)}{K * \left( \sum\limits_{t=1}^{N} \sum\limits_{i=1}^{3} \int_{\mathcal{O}_{\mathbf{x}}} u_{t;i}^q(\mathbf{x}) c_i^2 d\mathbf{x} \right)}.$$

Note that, for simplicity, we kept the values of the target intensity of each class (assuming the normalized data) fixed and equal to $\mathbf{c} = [0, \overline{\{I_t(\mathbf{x})\}}, 1]$. Model (1) was solved iteratively for 10 iterations, which was enough to ensure convergence. For analyzing influence of the smoothness of the bias field, we performed another experiment by progressively reducing the size of the truncated Gaussian kernel (Li et al., 2009) from 2048 to $1/2$ of this value, $1/3$, $1/4$, and so on. The results (not shown) indicated that starting from the standard deviation value of 2048/3 the results effectively did not change. Based on this observation, this value was selected for all further experiments.

Estimated bias fields for each of the five data sets exhibit high degree of resemblance, as illustrated in Figure 2. We have also considered simpler versions of the derived model (1) by setting one or two out of the three intensity correction factors ($\mathbf{a}$, $\mathbf{b}$, $B$) to their default values. Results of this experiment, illustrated in Figure 3 for the Jeffrey divergence, indicate that using sole bias correction significantly outperforms all other approaches and improves homogeneity of separate frames and similarity between the neighbouring ones.

### 3.2. Intra-set intensity standardization

In this experiment, we analyzed influence of the described bias correction and intensity scaling by simple histogram equalization (Kim and Paik, 2008) on training capability using a single data set. As the training data, we selected the 2929Q1 data set that has the largest amount of annotated data, both in terms of the number of frames and the number of mitochondria. The Jaccard index on the training data was used as the quality measure. Each training experiment was repeated twelve times and the distributions of the calculated results are shown in Figures 4 and 5.

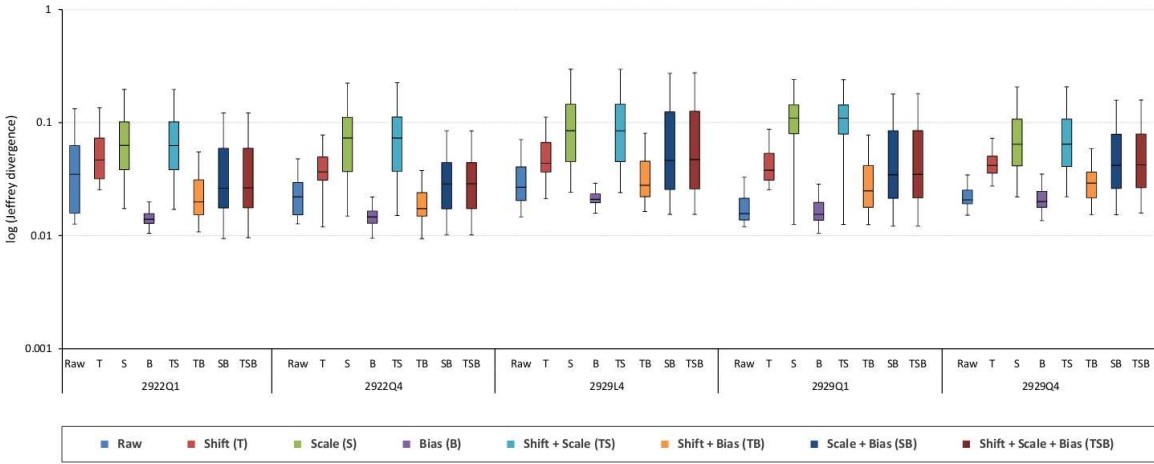

Figure 3: Results of intra-set intensity correction by different methods measured in terms of the Jeffrey divergence in the overlap regions. The value axis is presented in the logarithmic scale. Lower values indicate better performance. Here "B" is bias correction, and "S" and "T" denote modelled intensity scaling (slope) and shift (intercept), respectively. Whiskers of the boxplot indicate the maximum and the minimum value, respectively.

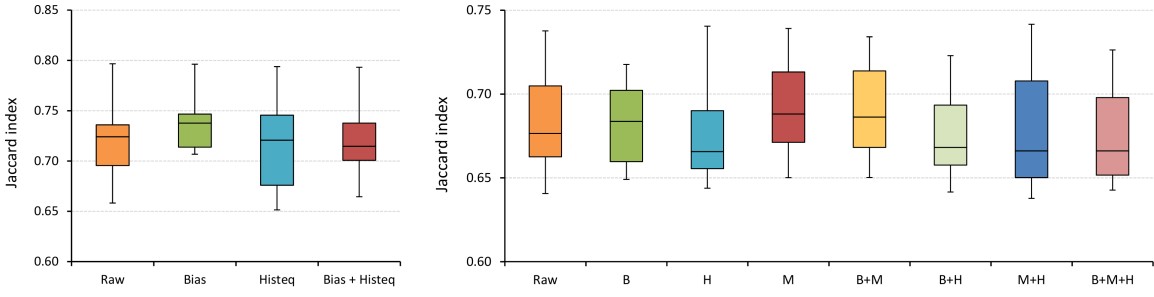

Figure 4: Distribution of the average Jaccard index on the test set for network trained on a single data set (2929Q1; left) and all five data sets combined (right). The network was trained twelve times on the data preprocessed by each method, every time with a different randomization seed for data splitting. Here "B" denotes bias correction, "H" — histogram equalization, and "M" — histogram mapping via exact histogram specification approach. Whiskers of the boxplot indicate the maximum and the minimum value, respectively.

### 3.3. Inter-set intensity standardization

Finally, we repeated the previous experiment on all the five data sets combined together. In addition to the previously described bias correction and histogram equalization, we also applied an exact histogram specification technique (Coltuc et al., 2006) to map the histogram of each data set to the corresponding histogram of 2929Q1. The intensity transformation curve was calculated for the entire set of frames belonging to the corresponding data set, and, consequently, intensity of each particular image was modified using the calculated mapping. Results of this experiment are shown in Figures 4 and 6.

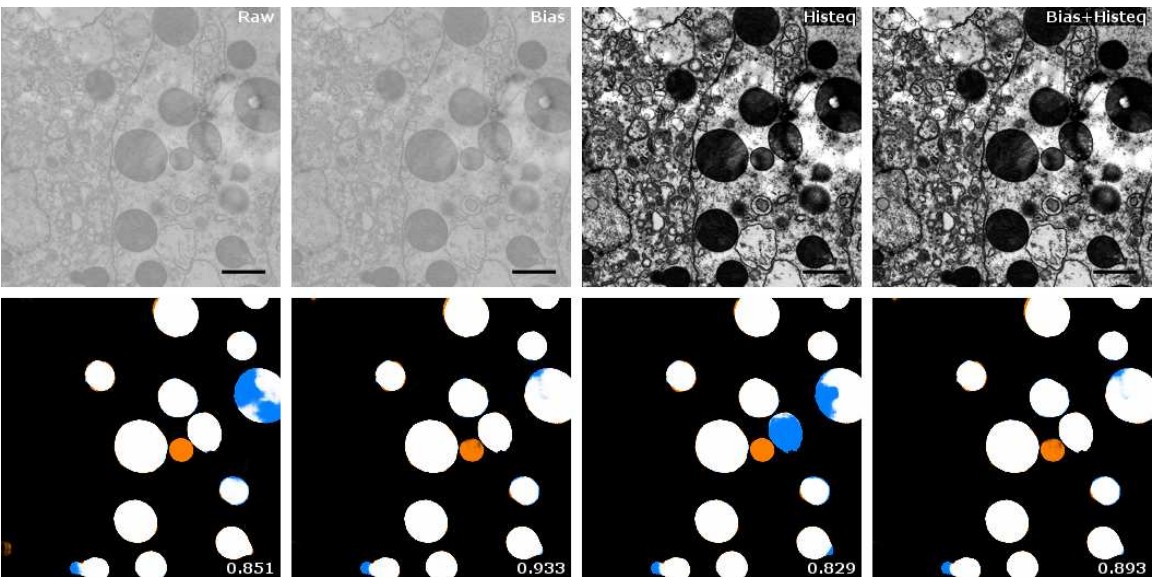

Figure 5: Representative example of the training results on 2922Q1 data set with significant performance improvement resulting from applied bias correction. The network was trained on the images preprocessed by different methods. In the bottom row, the ground truth and the actual segmentation are shown in complementary colors: blue and orange, respectively, such that the regions where they overlap appear white. The numbers indicate the corresponding value of the Jaccard index. The length of the scale bar is 1 μm.

## 4. Discussions and Conclusions

Problem of standardizing the data is of paramount importance for successful training of a neural network on data originating from different sources and is generally referred to as *domain adaptation*. Together with development of more robust training algorithms, it should enable successful application of deep learning approaches to large data arrays exhibiting high degree of diversity. Here we developed and analyzed methods for correcting and standardizing image intensity on the level of a single frame, a single data set and multiple data sets. We demonstrated, in particular, that developed bias correction approach (see Section 3.1) has positive effect on training results; see Figure 4(left) in Section 3.2. In this approach, the bias field is modelled as imperfection of the acquisition hardware that results in uneven illumination and is derived from the data by making a sole assumption about it being smooth. Quite surprisingly, a simple bias correction model, without additional intensity modification, outperformed all other approaches by far, as shown in Figure 3. An illustrative example of the benefits of performing the proposed bias correction is given in Figure 5.

Next, we considered the problem of standardizing image intensity for training a neural network on a combined data set consisting of multiple data sets; see Section 3.3. It is important to note that accounting for the difference in image intensity between training and test data sets can also be approached by augmenting the training data. In this work, we did not use this possibility as our goal was to investigate the impact of preprocessing

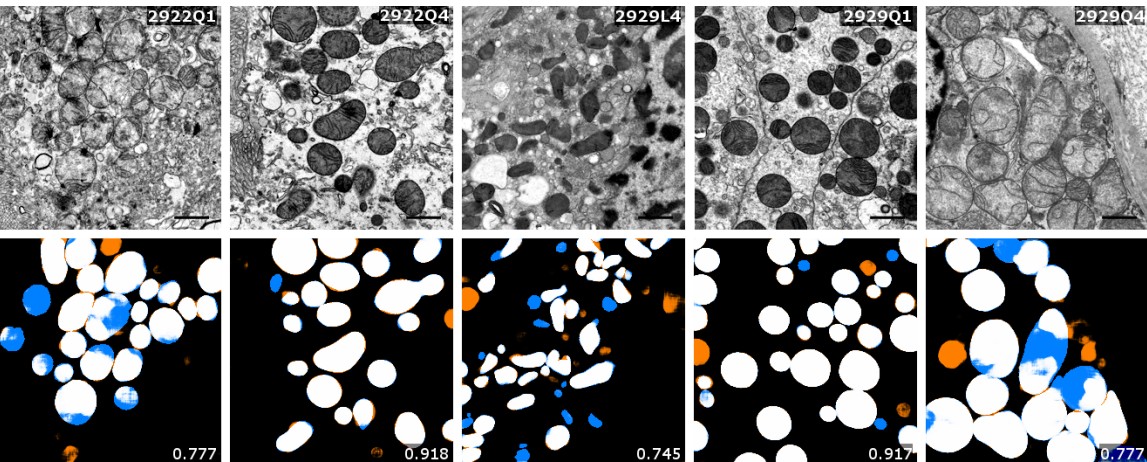

Figure 6: Representative examples of the training results on all five data sets combined. The network was trained on the images preprocessed by "B+M" method: bias correction with subsequent mapping using the exact histogram specification method. Contrast of the images (top row) was increased for visualization purposes. In the bottom row, the ground truth and the actual segmentation are shown in complementary colors: blue and orange, respectively, such that the regions where they overlap appear white. The numbers indicate the corresponding value of the Jaccard index. The length of the scale bar is 1 μm.

methods on the training performance. The results of this experiment are summarized in Figure 4(right). Several important conclusions can be drawn from this figure. First, training performance on the aggregate data set, consisting of all five data sets combined, is considerably lower than that on a single data set with sufficient amount of annotated data. Such performance decrease is explained by high data variability and minimization of this effect is the main goal of this research. Second, none of the analyzed methods resulted in clearly superior performance. Third, exact histogram specification mapping (Coltuc et al., 2006) produces the best overall results, with results of applying this method on raw and bias-corrected data being very similar. Fourth, applying the histogram equalization technique (Xiao et al., 2018), commonly used for this purpose, clearly deteriorates the results.

Figure 6 illustrates typical segmentation results on each of the five data sets on which our network was trained. Although the overall segmentation performance is quite good, this figure confirms that the difference in the amount of available training data per data set results in different performance; compare e.g. 2929Q1 to 2922Q1 and/or 2929Q4. Presence of orange-colored regions in some of the images in the bottom row also reflects inconsistencies in our ground-truth annotation as these objects were not annotated. As the next step in this project, we are planning to perform quality control on our manual annotations, which should further improve the segmentation results.

## Acknowledgments

We thank our colleagues Ian Alwayn, Asel Arykbaeva and Dorottya de Vries from the Department of Surgery (LUMC) for providing the tissue samples.

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
