# OpenReview forum: "Intensity Correction and Standardization for Electron Microscopy Data"
_MIDL.io/2021/Conference — MIDL 2021_

### Official Review · AnonReviewer2 · 2021-03-08

**Confidence:** 3
**Preliminary Rating:** 2
**Recommendation:** Poster
**Final Rating:** 3

**Summary:**

The authors present a method to correct the intensity variations in EM images. As a first step, the authors adapted the Coherent Local Intensity Clustering (CLIC) algorithm to clean these variations. Then they show, how intensity standardization can actually improve the segmentation performance on the EM data. They specifically showcase the mitochondria segmentation problem and show some promising results illustrating the positive effect of the proposed intensity standardization method.

**Strengths:**

- Data standardization is a key issue in medical image processing and machine learning. In this sense the tackled problem is important.
- Authors show promising results on sample EM data
- Authors compared different methods and tested segmentation model performance on all of them.

**Weaknesses:**

- The paper is very hard to follow.
- Technical novelty is not clear.
- The problem of interest is not well-illustrated so it is hard to understand for readers who are not much familiar with EM images.
- Even if the authors say that the dataset is very large, it is only composed of 5 stitches, which is limiting.
- Overall the presentation is not really good, figures are not illustrative, and therefore the paper is hard to follow

**Deanonymize Review:**

no

**Detailed Comments:**

- Overall the paper is very hard to follow. Throughout the text, the authors mention parameters (e.g. variance of the BIAS in section 3.1, 3 classes, etc...) that are optimized without explaining them or even introducing them. The structure of the experiment, data used for the training and test portion of the experiments, as well as the figures that are supposed to clarify the experiments are not clear.
- Authors used the CLIC algorithm to preprocess their images and then trained a segmentation neural network to extract the mitochondria areas in the images. Both the preprocessing and the segmentation portions are well-known techniques from the literature. The application of CLIC to EM seems to be novel. In this respect, the technical contribution of the paper, besides its application novelty is not clear.
- Authors state that they used 70/15/15 % portion of the "annotated frames" for training/validation/testing. The reviewer is not an expert in EM images but doesn't this introduce a bias during performance assessment as the frames coming from the same stitch are somehow correlated. Wouldn't it make more sense to prepare the partitions over stitches?
- It would be helpful if the authors can illustrate the intensity variation that they encounter with a few examples. Similarly, the statement at the first sentence of the 4th page may benefit from an illustrative figure. In its current version, it is not easy to fully understand.
- Figure 3 and 5 are not clear (details in the questions section)
- Conclusions presented in the last paragraph of section 4 are not clear.

**Final Rating Justification:**

Reviewer thank the authors for their detailed response. In light of the new details that the authors present and added to the manuscript, the reviewer updated their score.

**Justification Of The Preliminary Rating:**

The paper has a poor organization and presentation therefore it is hard to follow.
The authors used methods from the literature and applied them to their problem. in this sense the technical novelty is limited.
Results are not very strong.


**Paper Type:**

validation/application paper

**Questions To Address In The Rebuttal:**

- It would be interesting to check if most of the segmentation errors occur where the intensity variations are high.
- Authors should have illustrated the problem of interest in a figure. How do the intensity variations look like in TM images?
- Authors state that 3 expert clinicians selected ROIs for mitochondria labeling. Are all the ROI labeled by all experts? If so, how did the authors compose the consensus labeling? Or each expert picked ROIs independently and labeled the selected ROIs. In that case, how did the authors guarantee that the ROIs picked by the clinicians did not overlap?
- Authors state that they modified the CLIC algorithm to adapt it to EM images. From equation (1), it reads like the authors only tuned the parameters of the CLIC algorithm (a,b,B) for EM image. The technical novelty behind this parameter tuning is not clear.
- How do the authors decide to use 3 classes.
- Figure 3 is not clear. Comparing the left and the right graphs, the reviewer has the impression that adding more data to the training actually makes the results worse. Also, when all the 5 stitches are used for training (c.f.  caption of Figure 3) what is used for testing.
- Figure 5 is very crowded and without prior knowledge of the frames, the individualized results do carry much value. Can the authors be more specific about the message that they would like to relay with this figure?
- In section 3.3., why did the authors pick stitch 2929Q1 as the baseline and match the histograms of other stitches to this one. Wouldn't it make sense to also pick different stitches as the baseline for histogram matching?
- In section 4, last paragraph; it is not clear why -with the words of the author- "Presence of orange-colored regions in some images in the bottom row also reflects inconsistencies in our ground truth annotations..."? Can the selected ROI intersect? As the orange areas presented in the figure fall into the intersection areas (borders of the frames)  is there a chance that they may not be labeled in both images?
- How was the labeling conducted? Did the authors present the raw images to the expert readers or normalized images (either with the proposed method or another method)

**Special Issue:**

no

---

### Official Review · AnonReviewer3 · 2021-03-08

**Confidence:** 5
**Preliminary Rating:** 3
**Recommendation:** Oral
**Final Rating:** 3

**Summary:**

The submission explores different intensity normalization methods for TEM images and studies their effect when training a network to segment mitochondria using either data from the same stitch or from different ones.

Each TEM stitch is composed of different frames. Therefore, there is intra- and inter-variability among the frames with respect to the stitches. The authors test different intensity normalization methods to face both kinds of variabilities.

**Strengths:**

- The manuscript is well organized and most technical details are explained.
- The authors face a common problem in biomedical image analysis: data standardization. Not only the inter-sample variability affects the qualitative features of a biomedical image but also the calibration of the acquisition device or the sample preparation. Facing this kind of problem supports building larger datasets for training and most probably, improving the capacity to generalize of the trained models.
- All the experiments (model training) conducted in this work were replicated 12 times which ensures a fair comparison of the results.

**Weaknesses:**

I agree with the authors about the necessity of finding a way to combine annotated images in a way that network's performance can be improved. Especially for biomedical image analysis, a field in which annotated datasets are rather small compared to the ones in other fields of computer vision, the lack of these standardizing methods prevents from getting general enough models for image processing. On the other hand, deep neural networks are thought to be capable of learning and modeling those differences among images. The latter would be equivalent to the case in which the raw data is used to train the networks, which indeed did not prove to be worse than correcting for the inter-variability.
- Along with the manuscript, I missed a short discussion about these different approaches. Should we try to directly train networks that learn to standardize images? Or should we, on the contrary, put further effort into the normalization?

- While explaining the CLIC method, it is said that "Denoting the target intensity of each of the observed three classes as **c**...", where is this 3 coming from?

- It is not completely clear to me how T, S and B are obtained (Figure 2). Could it be possible to elaborate more on the text?

- The sum of the binary cross-entropy and the Jaccard index is used as a loss function. Is it weighted or is it the direct sum?

- In Figure 5, it can be seen clearly the effect of the data imbalance. In all the experiments, the images of 2929Q1 are better segmented than the belonging ones. This could be easily corrected by creating some image sampling during the training.


**Deanonymize Review:**

no

**Detailed Comments:**

- Figure 3, it would be recommendable to include the names of the abbreviations as done in Figure 2.

- " A contiguous 10 × 10 frames region was additionally selected from each of the annotated stitches, close to the location of the annotated ROI, for developing and testing the intensity-inhomogeneity correction algorithm". It is not clear why these regions are taken.
- In the discussion, in the first paragraph, please make it more clear that you are talking about intra-variability.
- "Data split was kept fixed within a single experiment in which different preprocessing methods were used." Please, reformulate the sentence as it is not clear.

- Figure 4 shows mitochondria along the edge of the images which could be potentially affected by the halo of the network. I would recommend using a different image for this Figure. Maybe, it could be the same one chose for Figure 6 so it is possible to see the differences between using all the data or only the stitch 2929Q1.

- If possible, I would redo Figure 5 by adding a colormap indicating the range of the Jaccard index values, and instead of showing 12 columns, each of them composed by 7 subcolumns, I would show 7 columns, with 12 subcolumns, so it is easier to discern between the preprocessing steps.

- In the discussion, it is said "Problem of standardizing the data is of paramount importance for successful training of a neural network on data originating from different sources. " I think it is also important to get versatile methods/models.

- Scale bars are strongly recommended in microscopy image analysis.




**Final Rating Justification:**

I thank the authors for their rebuttal. The manuscript has improved but there are still some technicalities that remain unclear.

**Justification Of The Preliminary Rating:**

I find the topic of the manuscript interesting for biomedical image analysis. However, I think the authors should address some technical comments written above to be certain about accepting their work.

**Paper Type:**

validation/application paper

**Questions To Address In The Rebuttal:**


- Is the Raw data scaled to the [0,1] range of intensity values?
- The images are downscaled for the training by a factor of ~10. Could this be the reason why so many little mitochondria are missed in the segmentation shown in Figure 6? What is the size range for the mitochondria? These details are important to chose your network architecture hyperparameters.

Further details about the data used in this work are missing.
- What is the pixel size of the images?
- What is the biological sample used? Are all the stitches coming from the same tissue and individual? Are the five different stitches coming from consecutive slices of the same tissue volume? This information is important to evaluate the real inter-variability.

As a future line, the work of the authors could be complemented if they could test the intensity correction and normalization approaches with images acquired with a different microscope or in a different lab. They could use some of the existing open-source datasets for mitochondria segmentation ([1] being the most recent one).


[1] MitoEM Challenge: Large-scale 3D Mitochondria Instance Segmentation, https://mitoem.grand-challenge.org/

**Special Issue:**

no

---

### Official Review · AnonReviewer1 · 2021-03-08

**Confidence:** 5
**Preliminary Rating:** 2
**Recommendation:** Poster

**Summary:**

The authors investigate the relevance of intensity standardization methods for the problem of mitochondria segmentation in transmission electron microscopy images. The authors use five different acquisition conditions to analyze the standardization methods. For each condition the authors acquire several fields of view that are then merged into a single image (so-called stitch). They then apply different methods to perform intra-stich intensity correction and standardization and inter-stitch intensity correction. For intra-stitch intensity correction they use CLIC (T,S,B), for intra-stitch intensity standardization they use histogram equalization (H) and finally, for inter-stitch intensity standardization they use an exact histogram specification technique (M). Results show that bias correction outperforms all other methods for a single stitch and that no correction seems the best option when training for all datasets.

**Strengths:**

-	Exhaustive experiments with respect to normalization methods
-	Good analysis of the results.
-	Interesting use of CLIC, although it should have been more clearly explained.
- Interesting problem to adress.

**Weaknesses:**

-	Lack of clarity. The use of the word stitch is not standard to refer to a collection of images tiled together. In general, the paper is hard to follow. The normalization methods (B+M, M, T, S, H, etc..) need to be dig out of the paper.
-	Lack of detail with respect to the image acquisition. Some acquisitions seem to be stained for mitochondria while other do not.
-	For completeness, the authors should describe the network used (Xiao 2018)
-	Lack of statistical analysis
-	Figure 5 is very hard to interpret, and shows that the results are heavily cross-validation dependent.
-	The authors refer to three classes in the text. It is the understanding of this reviewer that there are only two classes (mitochondria and background). Which is the third one?
-	The authors should have chosen a different field of view for Figure 4. The only targets are on the left and they are very hard to see.


**Deanonymize Review:**

no

**Justification Of The Preliminary Rating:**

The experimentation is good, and the problem of is of interest. I am missing more detail on the different datasets. Looking in detail at Fig. 1, it is clear that the authors present images with mitochondrial staining and without. This problem can not be addressed with simple image intensity standardization. That may be the reason why, when training a single network, no image intensity standardization outperforms the raw data. The problem at hand falls more on the realm of domain adaptation, which has not been discussed at all on the text.

**Paper Type:**

methodological development

**Special Issue:**

no

---

### Meta-Review · Area_Chair1 · 2021-03-29

**Recommendation:** Accept (Poster)

**Metareview:**

All reviewers agree that the idea of investigating different intensity normalization methods for mitochondria segmentation in EM images is of an interest to the MIDL community. There were major questions on several details of the proposed approach including experimental settings, but the authors have clarified most of them for reviewers to adjust their final ratings. While some of reviewer’s questions could be addressed better in the current manuscript, these are fixable in the final revisions.

**Paper Type:**

validation/application paper

---

### Decision · Program_Chairs · 2021-03-31

Accept